

# Stroboscopic aliasing in long-range interacting quantum systems

**Shane P. Kelly[1,2,3\*], Eddy Timmermans[4], Jamir Marino[3] and S.-W. Tsai[2]**

**1** Theoretical Division, Los Alamos National Laboratory, Los Alamos, New Mexico 87545, USA
**2** Department of Physics and Astronomy, University of California Riverside,
Riverside, California 92521, USA
**3** Institut für Physik, Johannes Gutenberg Universität Mainz, D-55099 Mainz, Germany
**4** XCP-5, XCP Division, Los Alamos National Laboratory, Los Alamos, New Mexico 87545, USA

⋆ shakelly@uni-mainz.de

## Abstract

We unveil a mechanism for generating oscillations with arbitrary multiplets of the period of a given external drive, in long-range interacting quantum many-particle spin systems. These oscillations break discrete time translation symmetry as in time crystals, but they are understood via two intertwined stroboscopic effects similar to the aliasing resulting from video taping a single fast rotating helicopter blade. The first effect is similar to a single blade appearing as multiple blades due to a frame rate that is in resonance with the frequency of the helicopter blades' rotation; the second is akin to the optical appearance of the helicopter blades moving in reverse direction. Analogously to other dynamically stabilized states in interacting quantum many-body systems, this stroboscopic aliasing is robust to detuning and excursions from a chosen set of driving parameters, and it offers a novel route for engineering dynamical $n$-tuplets in long-range quantum simulators, with potential applications to spin squeezing generation and entangled state preparation.

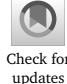

# 1 Introduction

The field of dynamical stabilization has a long tradition tracing back to the Kapitza pendulum in the mid 60s [1]: a rigid rod can be stabilized in an inverted position by parametrically driving its suspension point with a tuned oscillation amplitude and at high frequency. The working principle of a dynamically stabilized upside-down pendulum is the building block for realizing periodic motion in atomic physics, plasma physics and in the theory of dynamical control in cybernetical physics. Periodic drives are a versatile tool that can be employed to stabilize systems in configurations prohibited at equilibrium. Applications in the quantum domain range from cold atoms to trapped ions [2–9]: a drive with large amplitude and fast frequency can stabilize an entire band of excitations, turning the dynamics of a collective mode from a runaway trajectory into a periodic orbit. In this work, we propose a flexible route to engineer periodic dynamical responses characterized by arbitrary integer fractions of the period of the drive, relevant for a broad class of quantum many-body simulators.

Periodic dynamics in isolated many-particle systems, can be also found in the absence of an external drive. Examples range from quantum 'scars' [10–14] to the dynamical confinement of correlations [15–21] and encompass the role of dynamical symmetries [22–26] in evoking persistent temporal oscillations. The quest for time translation breaking in periodically driven

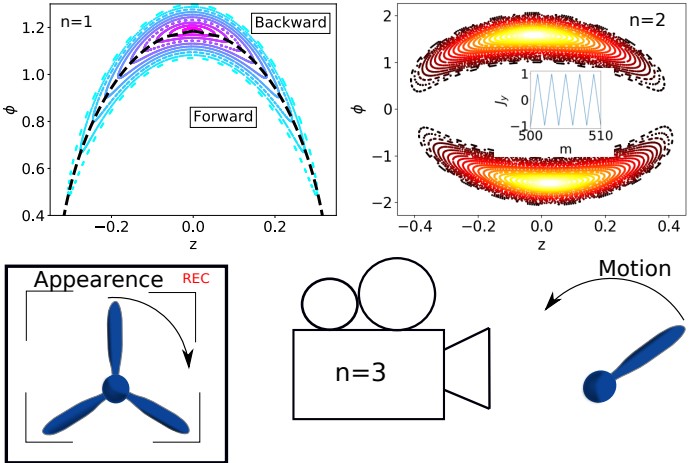

Figure 1: [Color Online] The top left figure shows the classical stroboscopic dynamics for an $n = 1$ resonance with $(t_1, t_2) = (2.1, 0.005)$. The black line shows the $H_1$ trajectory with period $\tau = nt_1$. In the region labeled "Forward" the stroboscopic dynamics appear to move forward along this trajectory (analogous behaviour holds for the region labeled "Backward"). This apparent reversal of motion is equivalent to the stroboscopic aliasing effect observed when the frame rate of a camera is faster than the rotation rate of a helicopter blade. The top right figure shows example of an $n = 2$ resonance with $(t_1, t_2) = (1.1, 0.05)$, and it contains an inset of the exact stroboscopic quantum dynamics that displays the $n = 2$ subharmonic response. The cartoon depicts an example of stroboscopic aliasing effects that occurs when the frame rate of the camera is $n = 3 - |\epsilon|$ times the rotation rate of the blade.

quantum systems [27–29] has recently morphed into the search for quantum time crystals [30–33]. A discrete time crystal (DTC) occurs when the discrete time translation symmetry of a periodically driven system is spontaneously broken into a smaller symmetry subgroup. One of the earliest identified examples of DTC occurs when a BEC bounces on a mirror that is driven at a resonance of a single particle trajectory [34–36] and illustrates the role of interactions in

breaking discrete time translation symmetry. In this example, a 1D BEC is trapped between an infinite barrier on the right and a linear ramp potential on the left. The linear ramp potential is driven at a frequency $\omega$ and the BEC oscillates at a frequency $\omega/2$ [34] by rolling up and down the ramp and bouncing off the wall. The now iconic example [37, 38] of DTC occurs when the spins of a disordered interacting spin chain are flipped at periodic intervals, and their local magnetization oscillates with a period twice the one of the spin flips. In this model, the stability of the time crystalline behaviour is provided by the extensive set of quasi-local integrals of motion which are characteristic of many-body localized phases occurring at strong disorder [39]. Since original experiments in trapped atomic ions and in nitrogen-vacancy centers [40, 41], many other mechanisms for time crystals have been proposed [42–53] and observed [54–58]. In all of these systems, the periodic dynamics are split into two parts: the natural dynamics of a system that possesses a $Z_n$ symmetry, and a kick process that sequentially switches among the $n$ symmetry sectors. An $n$-period DTC (or 'n-tuplets dynamics') occurs since it takes $n$ of such kick processes to bring the system back to its original configuration [59].

In this work we show how to engineer dynamics with arbitrary $n$-tuplets that are not distinguished by the sectors of a $Z_n$ symmetry. Differently from time crystals, their stability emerges as a cooperative effect between the natural dynamics and the kick process. Subharmonic response with any value of $n$ can be generated provided that the kick period is in resonance with the $n^{th}$ harmonic of a collective mode, and this collective mode remains stable, though deformed, during the kicked process. This results in stroboscopic dynamics which display period-$n$ oscillations between $n$ emergent dynamical fixed points.

By considering the kick akin to the sampling performed by a video camera, we identify this subharmonic response as similar to a type of stroboscopic aliasing that occurs when filming a single blade helicopter: when the helicopter blade is rotating at the $n^{th}$ subharmonic frequency of the camera's frame rate, its video will appear to have $n$ blades. Unlike the sampling performed by the camera, the kick acts on the many-body system increasing or decreasing the frequency of the system. This results in another stroboscopic aliasing effect in which the apparent $n$ stationary blade appear to slowly move forward or backwards depending on if the blade frequency was increased or decreased (cf. with Fig. 1). We show (in Section 2) that for a general class of kicks, both forward and backward aliasing appears and generates a set of $n$ stroboscopic fixed points that stabilize the subharmonic response. Stroboscopic aliasing produces also a set of $n$ unstable dynamical fixed points which we argue could be used for generating spin squeezing and entangled states.

Like the subharmonic response of the many body localized DTC, the stroboscopic aliasing subharmonic response is stable to quantum fluctuations and many body perturbations. Unlike the DTC, the subharmonic response discussed here is also stable to dissipation and strong interactions, both of which destroy many body localization and the stability it provides for the DTC. Finally, unlike previously identified subharmonic responses that are stable at large interactions [43], stroboscopic aliasing does not require a symmetry present in the dynamics. These aspects of the stroboscopic aliasing subharmonic response are discussed in Section 3.

## 2 Stroboscopic Aliasing

### 2.1 Model

We consider a long-range interacting Ising model [60–64] in which for $m$ cycles the unitary evolution operator, $U(m) = (U_1 U_2)^m$, is applied to the state of the system. Here $U_1$ and $U_2$ correspond to quantum evolutions at two different couplings strengths, resulting effectively in a periodic kick of the interactions. We define $U_a$ as a unitary generated by the following

hamiltonian

$$H_a = -\sum_{k=1}^{N} \sigma_k^x + \frac{\Lambda_a}{2N^{1-\alpha}} \sum_{k,j=1}^{N} \frac{\sigma_k^z \sigma_j^z}{|k-j|^\alpha}, \tag{1}$$

where $N$ is the number of spin-halfs, $\vec{\sigma}_k$, which live on a one dimensional lattice, and the unitaries are evolved for different times $t_1$ and $t_2$ and for different interaction strengths $\Lambda_1$ and $\Lambda_2$ (i.e. $U_a = e^{it_a H_a}$, with $a = 1, 2$). The Kac rescaling factor with $N^{1-\alpha}$ is to ensure the extensivity of the hamiltonian in the thermodynamic limit [65]. The subharmonic response emerges when $t_1$ is in resonance with a collective mode of $H_1$ and $t_2 \ll t_1$. Focusing our attention to this limit, we will refer to $U_2$ as the kick.

The emergent subharmonic response is most clearly explained in the $\alpha = 0$ infinite range limit in which the model reduces to the Lipkin-Meshkov-Glick (LMG) model [66–68]. In the large $N$ limit, dynamics reduce to the motion of the collective magnetization $J_\alpha = \frac{1}{N}\sum_i \sigma_i^\alpha$ [69]. In the LMG model, each of the $N$ spins interact with all the others with the same ferromagnetic coupling strength. Such permutational symmetry allows for efficient quantum many body simulations of the model, and admits an exact mean-field solution in the thermodynamic limit. Furthermore, the model represents an instance of solvable quantum phase transition in an all-to-all connected spin model, between a paramagnet and a ferromagnet, upon decreasing the value of the transverse field along $\hat{x}$ (cf. Eq. (1)) below a critical point. The exact mean-field solvability of the problem permits restricting dynamics to a few classical variables. This results from the all-to-all nature of the interaction combined with the classical nature of the collective spin when its length grows for $N \to \infty$. The phase space of the collective mode (or magnetisation) has conjugate variables given by $z$ (the projection of the spin onto the $z$ axis) and by the phase $\phi$ of the spin in the $x$-$y$ plane.

## 2.2  Subharmonic response

We now briefly recall the dynamical stability properties of such classical phase space, which will be relevant for the mechanism of stroboscopic aliasing at the centre of this work. The non-linear classical dynamics of $H_1$ are integrable and can display a separatrix for strong enough $\Lambda_1$. When $t_1$ and $t_2$ are large compared to the inverse of the spin coupling strengths, the classical dynamics has a chaotic structure in the same universality class as the standard map [70]. When $t_2$ is small, most of the integrable trajectories of $H_1$ remain unchanged except for when the kick frequency is in resonance with a harmonic of a trajectory of $H_1$; in this case, $t_1 \approx \tau/n$, where $\tau$ is the period of a trajectory of $H_1$.

When this condition is met for an integer $n > 1$, the dynamics display persistent subharmonic oscillations, and a few instances are shown in Fig. 1 and Fig. 2 (with $\Lambda_1 = 10$ and $\Lambda_2 = 0$). To understand why these oscillations occur and to assess their stability, we will first work in the limit $\Lambda_2 = 0$, and turn our attention to the first plot of Fig. 1 where we have shown a set of $U(m)$ stroboscopic trajectories near an emergent fixed point with a $n = 1$ resonance. There we have also plotted the resonant ($n = 1$) trajectory of $H_1$ in black. Since $t_1 = \tau(E)$, $U_1$ completes one period of the trajectory and evolves a spin initialized on this trajectory back to its initial point. Thus, ignoring for the moment $1/N$ quantum corrections (see [71] and the footnote [1]), we can approximate $U_1 \approx 1$ for initial states on this resonant trajectory. Similarly, when initial states start on an $H_1$ trajectory with period slightly less than $t_1$, they appear to move slightly forward along the trajectory by a time $t_1 - \tau$. Again, we can approximate $U_1(t_1) \approx U_1(t_1 - \tau)$ when $U_1$ acts in this region of phase space. Similarly when

---

[1] The commutation relations between the angular momentum components of the collective magnetization, vanish as $1/N$ for large $N$; when restored, they reintroduce subleading quantum effects on top of the classical motion of the collective mode.

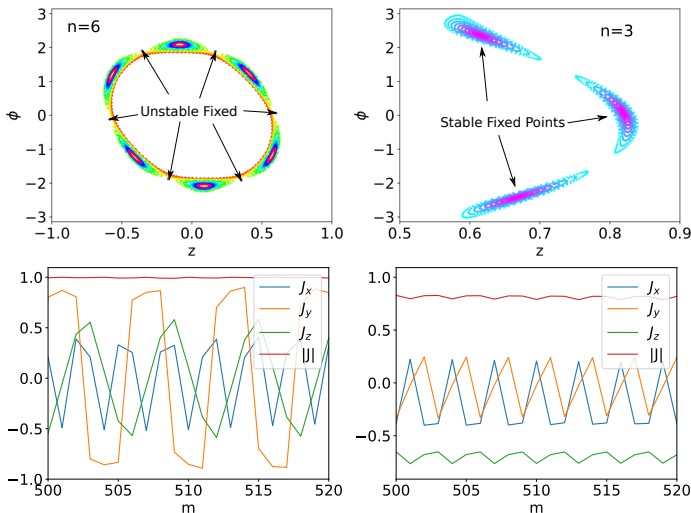

Figure 2: Stroboscopic classical Poincaré section (top) and exact stroboscopic quantum dynamics (bottom) for $N = 500$, $\Lambda_1 = 10$, $\Lambda_2 = 0$: with $(t_1, t_2) = (0.35, 0.2)$ (left), and $(0.3, 0.1)$ (right). The color (brightness) in the top plots distinguishes the initial state. The top plot depicts the emergent classical fixed points for $n = 6$ (left), and $n = 3$ (right). In the bottom plots, we show the $n = 6$ and $n = 3$ subharmonic oscillations due to $U_1$ moving between the different emergent fixed points. $|J|$ is plotted to illustrate that the fixed points stabilize the system against quantum dephasing. In the top left Poincaré plot, we have also highlighted the $n$ unstable fixed points that generically occur in addition to the stable fixed points.

$t_1 < \tau$, the state appears to move slightly backwards by a time $\tau - t_1$ and we can approximate $U_1(t_1) \approx U_1^\dagger(\tau - t_1)$. This inspires us to label the trajectories with $\tau < t_1$ as 'forward' trajectories and the trajectories with $\tau > t_1$ as 'backward' trajectories. This apparent forward and backward motion is the same stroboscopic aliasing effect that occurs when video taping a helicopter blade with a frame rate similar to the rotation frequency.

We now consider the action of the $U_2$ kick. For $\Lambda_2 = 0$, the kick is a $J_x$ rotation, and in the region of phase space shown in the first plot of Fig. 1, a $J_x$ rotation increases $z$ and keeps $\phi$ approximately constant. Therefore, when $z > 0$ a spin on a forward trajectory is kicked towards the backward trajectories, while when $z < 0$, a spin on a backwards trajectory is kicked towards the forward trajectories. Thus, in this region of phase space, the interplay of stroboscopic aliasing and the kick causes the spin to switch back and forth between the forward and backward trajectories and creates a new stroboscopic fixed point.

When the resonance condition occurs for $n > 1$ a similar description holds up to a few subtleties. First, $U_1$ only completes a fraction $(1/n)$ of a trajectory. Therefore, we should define the forward and backward trajectories based off the classical trajectories of the unitary, $U_1' = (U_1 U_2)^{n-1} U_1$. In the perturbative limit of small $t_2$, the classical periods and trajectories of $U_1'$ will only be slightly shifted from the LMG trajectories, and we can follow similar arguments as above. The dynamics defined by $U'(m) = (U_1' U_2)^m$ will then have a similar fixed point structure and trajectories as shown in Fig. 1, but will only capture the dynamics when looking every $n$ steps of $U$. Looking at every step, we see that $U$ will shift the fixed point and resonant trajectories of $U'$ to $n$ different $U'$ fixed points in phase space before returning to the original $U'$ fixed point. This shows that, at the resonances, there must be $n$ stroboscopic fixed points of the $U'$ dynamics, and this is confirmed in Fig. 2. Since these are fixed points of the $U'$ dynamics, the $U$ dynamics display a period-$n$ oscillation due to $U$ moving the spin between

the $n$ different fixed points of $U'$. In the analogy to stroboscopic aliasing, this subharmonic response is similar to a filmed single blade helicopter apparently showing multiple $n$ blades when the frame rate $1/t_1$ is $n$ times the frequency of the helicopter $1/\tau$.

In addition to the $n$ stroboscopic stable fixed points, a set of $n$ unstable fixed points also occur (See Fig. 2). These unstable fixed points separate the trajectories around the stable stroboscopic aliasing fixed points from each other and from the regular off-resonant dynamics. In the conclusion we speculate that these unstable fixed points could be used for squeezing and generation of entangled states.

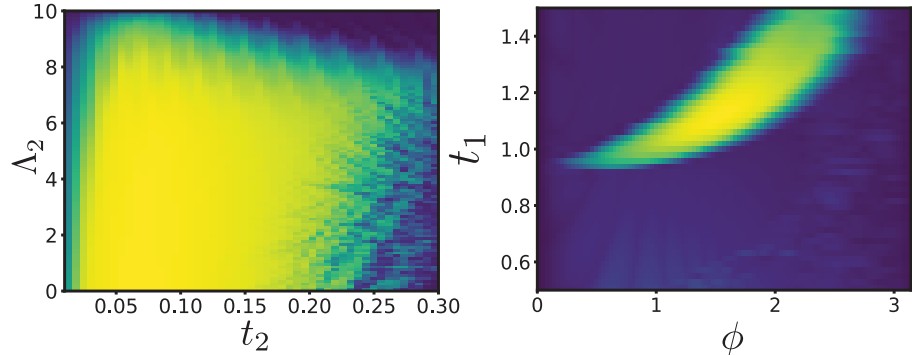

Figure 3: In this figure we demonstrate the stability of the $n = 2$ stroboscopic aliasing subharmonic oscillations to variation of hamiltonian parameters. The two panels are for $\alpha = 0$ and are computed using exact quantum dynamics. They show the order parameter $\max_f J_y(f)$ discussed in the text as a function of $\Lambda_2$, $t_2$ (left) and $t_1$ and the initial phase $\phi$ (right). In these plots, the brightest yellow corresponds to $J_y(f) = 1$, while the darkest blue to $J_y(f) = 0$.

## 3 Stability

### 3.1 Stability to quantum fluctuations

. Unlike the stroboscopic aliasing that occurs while filming helicopters, the stroboscopic aliasing subharmonic response is actively stabilized by the interplay between aliasing and kicking, and it does persist when the drive parameters are slightly detuned. First, we discuss the stability of stroboscopic aliasing to the accumulation of quantum fluctuations in the course of long-time dynamics. In the bare LMG model $H_1$, fluctuations lead to the collapse of periodic oscillations [72], while in the exact [73] numerical calculations, we find that such collapse does not occur for the aliasing subharmonic response. This can be understood in a semiclassical picture where quantum fluctuations are captured by a quantum diffusion process that spreads the wave function along the conservative classical trajectory [74]. Collapse of periodic oscillations occurs when the diffusion process reaches a steady state with the wave function completely spread out along the periodic trajectory performed by the classical dynamics.

For the stroboscopic aliasing subharmonic response, the steady state contains an oscillation that moves the spin between the $n$ dynamical fixed points. These oscillations remain quantum because the wave function remains localized around these fixed points. Qualitatively, this is expected by regarding quantum corrections as quantum jumps that move the spin off of its classical trajectory. In the large $N$ limit, these jumps are exponentially suppressed [74], and so they can only move a spin within the well of an emergent fixed point, but not between them.

Thus, we expect that quantum corrections cannot spread the state between the different stable emergent fixed points and that the subharmonic response to be robust to quantum fluctuations. This is confirmed by the stability of the subharmonic response after $m = 500$ oscillations, and the dynamics of $|J|^2 = \sum_\alpha \langle J_\alpha \rangle^2$, which shows that spins move along the surface of the Bloch sphere (See Fig. 2).

Therefore, one should expect the stroboscopic aliasing subharmonic response to be stable to variations in $t_a$ and $\Lambda_a$ as long as they only deform the emergent fixed point structure. To test the extent of this stability, we focus on the $n = 2$ case shown in Fig. 1 and work with an initial state completely polarized along the $J_y$ direction. As shown in the same figure, the subharmonic response is observed in oscillations of $J_y$ between 1 and −1. We therefore use the Fourier spectrum, $J_y(f) = \frac{1}{M} \sum_{n=1} e^{-ifn} J_y(n)$ of the $y$ component of the spin to asses the stability of the stroboscopic aliasing subharmonic response. When oscillations are stable for long times, the discrete Fourier spectrum, $J_y(f)$ will be singularly peaked around $f = \pi$. Thus, similar to [49], we take $\max_f J_y(f)$ as our order parameter for the $n = 2$ stroboscopic aliasing oscillation phase.

A phase diagram of this order parameter in the $t_2$ and $\Lambda_2$ parameter space is shown in Fig 3. The pronounced stability to variation in $\Lambda_2$ reflects the fact that any $U_2$ that connects the forward and backward trajectories in this region of phase space is sufficient to stabilize the fixed point there. When $t_2$ becomes large, the majority of the resonant trajectories around the fixed points become chaotic and the phase is destroyed. Fig 3 also shows that the phase is stable to variations in $t_1$. This is because there is a continuum of periods with $\tau = 2t_1$ which can be in resonance with $U_1$.

## 3.2 Stability to many body quantum fluctuations

Up to now, we have discussed the limit of $\alpha = 0$ in the hamiltonian (1). In this case, dynamics are well approximated by the motion of a single large spin, and the evolved states

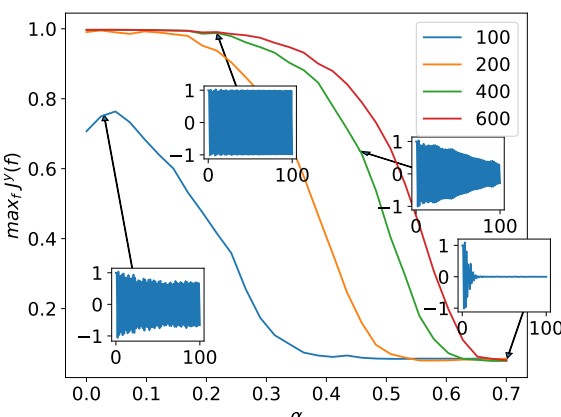

Figure 4: In this figure we demonstrate the stability of the $n = 2$ stroboscopic aliasing subharmonic oscillations to many body perturbations. The figure shows the order parameter $\max_f J_y(f)$ as a function of $\alpha$ for a few values of $N$. The insets show $J_y(t)$ at the points indicated by the arrows. Simulations are preformed using the DTWA and evolve an initial state polarized in the $\hat{y}$ direction under the hamiltonian (1) in one dimension. We evolve for $m = 500$ periods and compute $J_y(f)$ over this time window.

are constrained to a Hilbert space where the spins at different sites are indistinguishable by permutation symmetry. This Hilbert space has only $N$ states and does not fully reflect the many body nature of a realistic experiment. Furthermore, several exactly solvable mean-field limits can encounter severe modifications when higher-point cumulants can build as a result of genuine many-body interactions. This is of particular importance in view of assessing the stability of the novel dynamical regime discussed so far, and for extensions to trapped ions experiments, where the long-range nature of the spin-spin interactions is unavoidably present. We therefore study the robustness of the subharmonic response at finite $\alpha$. We use the Discrete Truncated Wigner Approximation (DTWA) which yields accurate results in long-range interacting models [75–81]. DTWA evolves the dynamics according to classical equations of motion, but treats exactly quantum fluctuations in the initial state by sampling over a discrete Wigner distribution [74].

We again compute $\max_f J_y(f)$ and the results are shown in Fig. 4. For $N = 100$, quantum diffusion occurs on observable time scales. As shown in the inset and discussed above for $\alpha = 0$, this decreases the amplitude of the subharmonic response but does not result in a complete decay. For $N = 200$, our numerics show that, up to computable time scales, the oscillations are almost perfect up to $\alpha = 0.2$ at which the subharmonic response starts to slowly decay. This indicates that for large values of $\alpha$, many body effects relax the oscillations before quantum diffusion in the collective Hilbert space occurs. As we increase $N$, this critical $\alpha$ grows to larger values indicating that these many body effects are a finite size effect and are suppressed at large $N$.

This result is consistent with previous results in driven long range interacting systems [9, 47, 63, 64, 82], and can be understood by a suppression of spin waves [9, 62–65, 83–88]. In the $\alpha = 0$ limit, spin waves are not excited due to permutation symmetry, while for finite $\alpha$, spins waves can be produced. In Ref. [64], they found that the generation of spin waves are suppressed by a factor small in $\Lambda_a / N^{1-\alpha}$ when $\alpha < 1$. This suggests that the $\alpha < 1$ dynamics are stable to many body perturbations for times up to $N^{1-\alpha}/(\max_a \Lambda_a)$, and that the stability of Stroboscopic aliasing increases with $N$ as shown in Fig 4. While the DTWA numerics cannot identify the critical value in the thermodynamic limit, they do show that oscillations are stable for finite $\alpha$, finite $N$ and within observable time scales. The stroboscopic aliasing is therefore not a fine tuned point in parameter space, rather, it shows robustness to the inclusion of long-range spin-spin interactions, relevant in trapped ions implementations. In this respect, it survives, for times accessible to DTWA, a purely mean-field description of dynamics.

## 3.3 Generality and stability to dissipation

. We believe that the stroboscopic aliasing subharmonic response discussed in this work is a general phenomenon provided a few requirements are satisfied. The collective mode should have only one dominant frequency, otherwise the kick cannot be in resonance with a single period. Furthermore, the kick must deform the collective mode, although not completely destroy it. The trajectory of the deformed collective mode should cross the bare trajectory in two points since this will allow for the dynamics of $U_1' = (U_1 U_2)^{n-1} U_1$ to cross back and forth across the resonant trajectory. Notice that these requirements are easily satisfied when the classical phase space of the collective mode is two dimensional because this guarantees regular trajectories with only one frequency. Despite such required regularity in the collective mode dynamics, integrability is not required as demonstrated by the robustness of the subharmonic response to many body perturbations at finite $\alpha$.

Furthermore, the dynamics of the collective mode are not required to be conservative either. We demonstrate this aspect by considering the effect of a global spin decay, which occurs naturally in cavity QED experiments [89–92]. The effect of global spin decay is modeled via

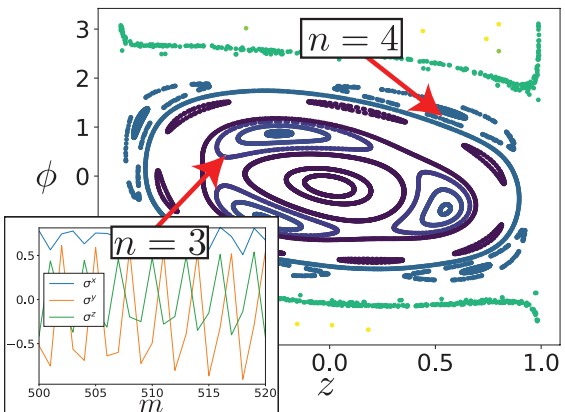

Figure 5: Stroboscopic aliasing subharmonic response in the presence of collective spin emission. Depending on initial conditions an $n = 4$ or an $n = 3$ subharmonic oscillation can occur. The inset also shows the $n = 3$ subharmonic response for an initial state with $z = 0.5$ and $\phi = -0.5$. In this figure $\alpha = 0$, and we have used a mean field approximation which assumes the thermodynamic limit: $N \to \infty$.

Lindblad evolution:

$$\partial_t O = i [H_a, O] + \kappa \left( 2J^+ O J^- - \{J^+ J^-, O\} \right), \tag{2}$$

where the Lindblad jump operators, $J^{\pm}$, are the total spin raising and lowering operators, and $O$ is an arbitrary operator evolving in the Heisenberg picture. To solve the dynamics for the collective observables, $\langle J_\alpha \rangle$, we make a mean field approximation, $\langle O_1(t) O_2(t) \rangle = \langle O_1(t) \rangle \langle O_2(t) \rangle$, to truncate the hierarchy of equations generated by Eq. 2 and numerically solve the closed set of equations for $\partial_t \langle J_\alpha \rangle$. Such an approximation is valid for times large [45] in $N$, and therefore capture the thermodynamic limit $N \to \infty$.

Stroboscopic aliasing occurs when the dissipative dynamics has a limit cycle. This occurs at $\kappa = 0.5$ and for initial states polarized close to $\langle J_x \rangle = -1$ [45]. Choosing $t_1$ to be in resonance with the period of these collective limit cycles, we are able to find a subharmonic response and have plotted examples for $n = 4$ and $n = 3$ in Fig. 5, at $\alpha = 0$. In the same plot, we show an example of a persistent subharmonic response for an initial state initialized with $z = 0.5$ and $\phi = -0.5$. The oscillation is periodic after every three kicks as it moves between the $n = 3$ fixed points shown in the Poincaré section.

The robustness of the phenomenon to coupling to a bath, is of importance in view of a growing interest in the time crystal community towards dissipative limit cycles and period doubling phenomena in quantum optics related platforms [44, 45, 47, 93–105]. A thorough study of the interplay of noise and interactions lies beyond the scope of this work, but the resilience of subharmonic dynamics shown in Fig. 5 is encouraging in the perspective of realising stroboscopic aliasing in experiments.

## 4 Conclusion

To conclude, we remark that the stroboscopic aliasing effects discussed so far should be observable in experiments. The hamiltonian (1) is used to describe trapped ion experiments [106, 107] in which the transverse field is easily controlled and can be employed to implement the kicks of $\Lambda_i$. Furthermore, the emergent unstable fixed points could also be used to create

squeezing or more general entangled states in a way similar to the bare unstable fixed points of $H_1$. Similar to Refs. [69, 108–110] such fixed points have two stable directions and two unstable directions. A quantum state initialized on the unstable fixed point, compresses in the two stable directions and expands in the two unstable direction creating, on short times, a squeezed state. At longer times, the state is stretched further apart and no longer resembles a squeezed state, yet it might show non-gaussian entanglement with properties controlled by the shape of the separatrix [109]. Since separatrices in the stroboscopic aliasing discussed here, have different topologies, they can open opportunities to generate new classes of entangled states in trapped ions simulators or in ultracold atoms experiments [69, 110], potentially with novel metrological uses. Finally, studying the critical properties of the transition away from the stroboscopic aliasing response, and analyzing its interplay with quantum fluctuations [111, 112] remains an interesting future direction of research. After completing of this work, we became aware of [82], which finds a similar subharmonic response to the stroboscopic aliasing discussed above, but with the drive on the transverse field and with time dependence $sin(2\pi t)$ instead of a kick.

# Acknowledgements

S. P. K. would like to acknowledge stimulating discussions with Levent Subasi and David Campbell. S. P. K. acknowledges financial support from the UC Office of the President through the UC Laboratory Fees Research Program, Award Number LGF-17- 476883. S. P. K. and J. M. acknowledge support by the Dynamics and Topology Centre funded by the State of Rhineland Palatinate, and the Deutsche Forschungsgemeinschaft (DFG, German Research Foundation) – Project-ID 429529648 – TRR 306 QuCoLiMa ("Quantum Cooperativity of Light and Matter"). S. W. T. acknowledge support by National Science Foundation (NSF) RAISE TAQS (award no. 1839153). The research of E. T. in the work presented in this manuscript was supported by the Laboratory Directed Research and Development program of Los Alamos National Laboratory under project number 20180045DR.

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
