# Peer review of "Stroboscopic aliasing in long-range interacting quantum systems"

_SciPost Physics Core, doi:SciPost Phys. Core 4, 021 (2021)_

## Round 1 · Referee Report · Anonymous (Referee 1) · 2021-7-7

Strengths
2)Interesting results (subharmonic response with arbitrary multiple of the drive frequency)
3)Intuitive and qualitative picture to understand the main results (discussion in Sec. 2.2 for example, analogy with aliasing)
Weaknesses
1)The discussion of the role of quantum fluctuations due to finite-range interactions is limited to Sec 3.2 and would probably benefit from some revision.
2)Same for the discussion on the role of dissipation
Report
Furthermore the authors provide a nice interpretation of this phenomenon in terms of stroboscopic aliasing.
In practice the authors focus on a long-ranged Ising chain with a transverse field, with the strength of the Ising interaction which is modulated, alternating two values within one single period. The authors mostly discuss the fully connected case (alpha=0) for which one can reduce the dynamics to classical equations of motion. In this limit the authors provide a nice and transparent physical interpretation for the origin of the aliasing. In Sections 3.2 and 3.3 they discuss the role of quantum fluctuations due to finite-range interactions and dissipation.
Both these sections would require in my opinion some change both in terms of content and of presentation of the results(see below)
Overall, I think the paper meets the criteria for publication in Scipost and I would be happy to recommend publication provided the authors address the points below.
Requested changes
Sec 3.2
I am puzzled by the results in Fig.4 and by the conclusions the authors draw from them. In particular the fact that at finite alpha it seems that increasing N the stability of the n=2 aliasing increases, leading to the conclusion in Sec. 3.2 that:
"The stroboscopic aliasing is therefore not a fine tuned point in parameter space, rather, it shows robustness to the inclusion of long-range spin-spin interactions....In this respect, it survives a purely mean-field description of dynamics."
I would have expected that for alpha\neq0, when the model is non-integrable, the system would heat up towards infinite temperature in the thermodynamic limit as expected from a periodically driven system.
Could the authors comment on this expectation and whether it should be met by their model at finite alpha?Can they quantify this heating?
In this light, the fact that increasing the system size makes the aliasing more and more stable, as shown in Fig.5 is, according to me, a demonstration of the fact that the method chosen by the author (DTWA) is not able to actually capture thermalization. It seems instead that the fluctuations included by DTWA become less and less important in the thermodynamic limit. Can the authors comment on this point?
Overall I am rather skeptical about the stability of the aliasing to many-body quantum fluctuations since, as the authors state at the beginning of Sec.3 .3, it requires the existence of a well defined single-frequency collective mode, which is probably not the case in a generic many-body setting. Can the authors comment on this point?
Sec. 3.3
The authors could improve the presentation of this section, including (i)clarifying which case they are considering in Fig 5 (fully connected alpha=0 or not?Finite N or large N), (ii) writing down the Lindblad master equation they consider and (iii) mention in few words how they actually solve it.
Minor Issue:
References - some of the articles seem misplaced or not properly cited. For example Ref 39 has, from what I can see, nothing to do with Discrete Time Crystals and should not be cited together with 37-38.

Author: Shane Kelly on 2021-08-20 [id 1695]
(in reply to Report 1 on 2021-07-07)Response is in attached PDF file.
Attachment:
response_Aliasing.pdf

---

## Round 2 · Author Response

Dear Editor and Referee,
Thank you for sending us the correspondence on our manuscript “Stroboscopic aliasing in
long-range-interacting quantum systems” scipost 202103 00022v1. We thank the referee
for their positive comments and constructive suggestions. We have made all the suggested changes and marked them in blue on the manuscript. We hope that, in the current form, our work will be found suitable for publication in Scipost.
Yours Sincerely,
Shane P. Kelly, Eddy Timmermans, Jamir Marino, Shan-Wen Tsai

---

## Round 2 · List of Changes

Major changes are highlighted in blue and are discussed in the response to the referees comments and suggestions.

---

## Editorial Decision

published